# The Content of Phenolic Acids and Flavonols in the Leaves of Nine Varieties of Sweet Potatoes (*Ipomoea batatas* L.) Depending on Their Development, Grown in Central Europe

**DOI:** 10.3390/molecules25153473

**Published:** 2020-07-30

**Authors:** Barbara Krochmal-Marczak, Tomasz Cebulak, Ireneusz Kapusta, Jan Oszmiański, Joanna Kaszuba, Natalia Żurek

**Affiliations:** 1Department of Plant Production and Food Safety, Carpathian State College in Krosno, 38-400 Krosno, Poland; 2Department of Food Technology and Human Nutrition, Institute of Food Technology and Nutrition, College of Natural Sciences, University of Rzeszów, 35-601 Rzeszów, Poland; tomcebulak@gmail.com (T.C.); ikapusta@ur.edu.pl (I.K.); jkaszuba@ur.edu.pl (J.K.); nataliazurek7@gmail.com (N.Ż.); 3Department of Fruit, Vegetable and Plant Nutraceutical Technology, Faculty of Biotechnology and Food Science, Wrocław University of Environmental and Life Science, 51-630 Wrocław, Poland; jan.oszmianski@up.wroc.pl

**Keywords:** sweet potatoes, leaves, varieties, BBCH, phenolic acids, flavonols, ABTS, DPPH, FRA, antioxidant activity

## Abstract

The aim of the study was the qualitative and quantitative analysis of the bioactive components present in the leaves of 9 sweet potato cultivars grown in the moderate climate in Poland, which were harvested at different growth stages according to the BBCH (Biologische Bundesanstalt, Bundessortenamt und Chemische Industrie) scale (14, 51, 89). It was found that sweet potato leaves contained 7 polyphenolic compounds, including 5 chlorogenic acids—neochlorogenic acid (5-CQA), chlorogenic acid (3-CQA), 4-cryptochlorogenic acid (4-CQA), 34-di-*O*-caffeoylqunic acid (3,4-CQA), 3,5-di-*O*-caffeoylqunic acid (3,5-CQA)—and 2 flavonoids, quercetin-3-*O*-galactoside (Q-3-GA) and quercetin-3-*O*-glucoside (Q-3-GL). Their content depended on the genotype of the examined cultivars and on the stage of leaf development. The mean content of the identified polyphenolic compounds in the examined cultivars ranged from 148.2 to 14.038.6 mg/100 g^−1^ DM for the leaves harvested at growth stage 14 according to the BBCH scale. In the case of leaves harvested at BBCH stage 51, the concentration of polyphenolic compounds ranged from 144.76 to 5026.8 mg/100 g^−1^ DM and at BBCH stage 89 from 4078.1 to 11.183.5 mg/100 g^−1^ DM. The leaves of the Carmen Rubin cultivar collected at stage 14 contained the highest amount of polyphenolic compounds, while Okinava leaves had the highest amount of these compounds at stage 51. The highest content of polyphenolic compounds in leaves at BBCH growth stage 89 was found in the Radiosa variety. The highest concentration levels were found for 3-CQA at all stages of leaf development. Significant correlations between polyphenol content and antioxidant activity measured by 2,2’-azino-bis(3-ethylbenzothiazoline-6-sulfonic acid (ABTS), 2,2-diphenyl-1-picrylhydrazyl (DPPH), and ferric reducing/antioxidant power (FRAP) were found. The results of this experiment revealed that the growth stages and genetic properties of cultivars have a very significant influence on the content of phenolic acids and flavonols in sweet potato leaves. The results are innovative and can have a practical application, as the knowledge of the content of the substances under study makes it possible to determine the optimal management practice of sweet potato leaf harvest in order to obtain more top-quality raw material.

## 1. Introduction

Climate changes in Central Europe caused by increasing temperatures and decreasing precipitation can result in a loss or decrease of the significance of the presently cultivated plant species; therefore, it seems necessary to start growing alternative plants from the tropical climate and adapt them to new climate and soil conditions [1]. Such a species can be sweet potato (*Ipomoea batatas* L.), which is a plant from the tropical and subtropical climate whose usable parts are tubers and leaves. Sweet potato belongs to high-yield vegetable plants [2]. On a global scale, it ranks fifth after the cultivation of rice, wheat, corn, and cassava [3]. It is grown in all subtropical countries in America, Africa, and Asia, and it is the most popular of all tropical and subtropical tubers in the world [4]. In terms of human consumption, it is the most important edible plant in the world, in particular in Africa, parts of Asia, and the Pacific Islands. In suitable conditions, it can also be cultivated in the temperate zone [5,6]. The usable parts are underground tubers and leaves, which—in various parts of the world—are the raw material for the preparation of soups, teas, and salads, as well as grown as a raw material of the pharmaceutical and food industries [7]. Research by Dinu et al. [8] shows that sweet potato leaves have the highest antioxidant potential among leafy green vegetables. Research by Ishida et al. [9] and Sun et al. [10] showed the high nutritional value of sweet potato leaves expressed as the amount of minerals (Ca, P, Mg, Na, K, Fe, Cu, Zn), soluble fibre, vitamins (C, E, B1, B2, B6, niacin, biotin, carotene, pantothenic acid), and protein. The content of all of the above-mentioned components was much higher than in sweet potato tubers and stems, whereas the INQ (the Index of Nutrition Quality illustrating nutrient density) for protein, fibre, and most minerals was above 2 (an index of 1 means a well-balanced product in terms of the analysed ingredient). Johnson and Pace [11] studied sweet potato leaves and found a high level of unsaturated fatty acids, mainly linolenic and α-linolenic acid. According to research by Neel and Fant [12], sweet potato leaves demonstrate high biological activity expressed in high antioxidant activity and the ability to compensate for vitamin A deficiency. According to a number of authors, with a high content of polyphenols and vitamins C, B1, B2, and B6, sweet potato leaves help scavenge free radicals, and they have anti-ageing, anti-cancer, antibacterial, anti-inflammatory, antihypertensive and antidiabetic (type 2 diabetes) properties [13,14,15,16]. In their research, Kurata et al. [17] showed that polyphenolic compounds isolated from sweet potato leaves are very effective in reducing the increase of gastric cancer cells (Kato III), colorectal cancer cells (DLD-1), and leukaemia cells (HL-60). In addition, sweet potato leaves are rich in lutein, which is used to prevent macular degeneration [18]. However, the health benefits of plant raw materials depend to a large extent on the level of bioactive ingredients, which in turn depend on plant adaptation to the environmental conditions during the establishment, growth stages, and genetic properties of cultivars [7]. Therefore, the aim of this paper was to determine the influence of leaf growth stages and cultivar genotype on the content of phenolic acids and flavonols properties in the leaves of nine sweet potato cultivars grown in central Europe.

## 2. Results and Discussions

### 2.1. Identification of Phenolic Compounds in Sweet Potato Leaves

The identification of phenolic acids and flavonoids was based on the characteristics of ultraviolet spectra, mass spectra (UPLC)-PDA-MS/MS Waters ACQUITY system (Waters, Milford, MA, USA) and benchmark substances (Table 1) [19,20,21]. The polyphenolic compound profile was determined on the basis of UV-VIS adsorption maxima spectra, *m*/*z* mass-to-charge ratio, and fragmentation ions achieved from collision-induced dissociation (CID). The results are presented in Table 1.

In total, 7 polyphenolic compounds from two categories were identified: phenolic acids and flavonols. Within the first category, 5 hydroxycinnamic-acid derivatives were found. Peaks 1, 2, and 3 were characterised by the same *m/z* value of [M−H]^−^
*m*/*z* 353, whereas the fragmentation ion spectrum included two characteristic fragments with a mass of 179 and 191 respectively, pointing to the presence of caffeic and quinic acid radicals in the particle. The compounds were identified as esters of caffeoylquinic acid: 3-CQA, 5-CQA, and 4-CQA, which are usually called chlorogenic acid (Rt = 2.335), neochlorogenic acid (Rt = 2.960), and 4-cryptochlorogenic acid (Rt = 3.139). Peaks 6 and 7 had a pseudo molecular *m*/*z* peak of 515, which was 162 units larger than caffeoylquinic esters. The registered mass loss due to the parent peak fragmentation indicated the presence of additional caffeic acid radicals. The compounds were identified as dicaffeoylquinic esters: 3,4-diCQA and 3,5-diCQA, respectively. The other peaks 4 and 5 belonged to the flavonol group. Those peaks were characterised by the same pseudo molecular *m/z* peak of 463. The conducted fragmentation revealed a fragmentation ion 301 that is characteristic for quercetin, resulting from the formation of fragment 162, which is characteristic for a hexose particle. Therefore, peaks 4 and 5 were categorised as quercetin-3-*O*-galactoside and quercetin-3-*O*-glucoside, respectively.

### 2.2. Antioxidant Activity of Sweet Potato Leaves Depending on BBCH Growth Stage

According to a number of authors, the climate conditions during vegetation determine the intensity of abiotic stress factors (UV radiation, temperature, precipitation), which are the main factor influencing the synthesis of secondary metabolites, whose level depends on the vegetative parts of the plant and its growth stage [22,23]. According to Padda and Picha [24], the most secondary metabolites are synthesised in the leaf blade. Research by Jang and Koh [25] and Hue et al. [26] showed that another factor differentiating the antioxidant activity is the genetic variability of cultivars. The antioxidant activity of sweet potato leaf extracts depending on the BBCH (Biologische Bundesanstalt, Bundessortenamt und Chemische Industrie) growth stage measured with 2,2’-azino-bis(3-ethylbenzothiazoline-6-sulfonic acid (ABTS ^o+^), 2,2-diphenyl-1-picrylhydrazyl (DPPH), and ferric reducing/antioxidant power (FRAP) assays is listed in Table 2.

The results show that the variability range of antioxidant activity depends on the cultivar and leaf growth stage. In the first harvest date at BBCH stage 14, the highest antioxidant activity in ABTS ^o+^, DPPH, and FRAP assays was found in the leaves of the Okinava sweet potato and the lowest was found in the White Triumph cultivar. The decrease of antioxidant activity of leaves among sweet potato cultivars, from the highest to the lowest, was the following: Okinava > Jewel > Carmen Rubin > Radiosa > Beauregard > Purple > Molokai > Satsumo Imo > White Triumph. Meanwhile, at this stage of leaf harvest, there were significant correlations between the applied ABTS ^o+^, DPPH, and FRAP assays and total polyphenolic compounds (Table 2). The antioxidant potential of leaves harvested at BBCH stage 51 was ordered in the following way: Radiosa > Beauregard > Purple > Okinava > Jewel > Carmen Rubin > White Triumph > Satsumo Imo > Malokai. In the FRAP assay, the antioxidant activity ranged from 617.43 µmol TE (Trolox)/100 g DM in the Radiosa leaf extract to 28.47 µmol TE (Trolox)/100 g DM in the Purple leaf extract. In this case, high correlation coefficients were also found between the applied methods of testing the antioxidant activity and the total polyphenolic content. Only for the Beauregard cultivar did the correlation coefficients between antioxidant activity assays (ABTS ^o+^, DPPH, FRAP) and general polyphenols turn out to be insignificant (Pearson correlation R2 = −0.395, R2 = −0.369, R2 = −0.381, respectively). Antioxidant activity assays of extracts of leaves harvested at BBCH stage 89 showed the highest and the lowest antioxidant activity in the leaves of the Radiosa and Satsumo Imo cultivars, respectively. Xu et al. [27] tested 116 cultivars of sweet potato and found that the total share of phenolic acids amounted to 79.6% of the total polyphenolic content. According to Padda and Picha [24], the antioxidant activity of sweet potato leaves is differentiated by growth stages. In their research, the authors achieved the highest antioxidant activity in young unripe leaves of sweet potato, with antioxidant potential expressed as Trolox equivalents measured with the DPPH method amounting to an average of 99.61 mg g^−1^ DM. The authors also demonstrated that the correlation coefficient between antioxidant activity and total polyphenolic content amounted to 0.98. A similar high correlation coefficient of R2 = 0.98 between antioxidant activity measured with the ABTS and DPPH method was arrived at by Zhang et al. [21] and Fu et al. [28]. On the other hand, Jeng et al. [29] studied sweet potato leaves and observed a tendency toward the age-dependent decreasing antioxidant ability of leaves, and the tendency was confirmed in the conducted antioxidant activity assays: DPPH and FRAP. However, our own research points to something opposite: the highest average antioxidant activity measured in ABTS, DPPH, and FRAP assays was found in the oldest leaves harvested at BBCH stage 89, i.e., at full ripeness. Different results were achieved in a research study by Liao et al. [30], who found that the antioxidant activity level of sweet potato leaves was more correlated with the flavonol content than with the general polyphenolic content. Research conducted by Fidrianny et al. [31] showed that the antioxidant activity results of sweet potato leaves are influenced by extraction methods. Their results indicate that alcohol extracts show the highest antioxidant activity expressed in ABTS and DPPH assays.

### 2.3. The Polyphenolic Content in Sweet Potato Leaves Depending on the Leaf Growth Stage and Cultivar

The polyphenolic content in the leaves of the analysed cultivars is presented in Table 3 and Figure 1. The total polyphenolic content was significantly differentiated depending on the cultivar and leaf growth stage. The average content of the identified polyphenolic compounds in the cultivars under study ranged from 148.2 to 14038.6 mg/100 g^−1^ DM for leaves harvested at BBCH stage 14.

In the case of leaves harvested at BBCH stage 51, the polyphenolic compound concentration ranged from 351.35 to 5026 mg/100 g^−1^ DM, whereas at BBCH stage 89, the results varied from 4802.4 to 11.183.3 mg/100 g^−1^ DM. A summary of the results is presented in Table 3, and a graphic interpretation after statistical processing is shown in Figure 1, Figure 2, Figure 3 and Figure 4. Our own identification of phenolic acids was corroborated by the research [19,23,25,29]. In the research by Jung et al. [32], the dominating compound in sweet potato leaves was 5-neochlorogenic acid (5-CQA), and its amount depended on the genetic properties of cultivars. On the other hand, our own research results point to a significant dominance of chlorogenic acid (3-CQA) in the analysed leaves, in particular those harvested in the last harvest round (BBCH 89). Such results can be caused by different climate and soil conditions during the cultivation of sweet potato in Poland. The concentration of chlorogenic acid (3-CQA) in the analysed samples ranged from 37.38% in Satsumo Imo leaves harvested at BBCH stage 51 to 69.83% in Jewel leaves harvested at BBCH growth stage 89 (Table 3). Li et al. [18] studied phenolic acids in the leaves of 14 sweet potato cultivars and also identified phenolic acids that were not identified in our own research, that is: caffeic acid, ferulic acid, and 4,5-di-CQA. Their results indicate that the total polyphenolic content in leaves ranged from 2640.2 to 4200.9 mg/100 g^−1^ DM, whereas the identified levels of phenolic acids in sweet potato leaves from their own cultivation depended on the cultivar and leaf harvest date and ranged from 13.113 mg/100 g^−1^ DM in Okinava leaves harvested at BBCH stage 14 to 148.2 mg/100 g^−1^ in White Triumph leaves harvested at the same growth stage. In turn, Sasaki et al. [33] studied the phenolic acid content in leaves of 6 cultivars and 2 lines of sweet potato harvested every month from May to August and obtained the highest concentration of 3,5-di-CQA. Similarly, Truong et al. [23] studied the leaves of 9 sweet potato cultivars and found the highest concentration of 3,5-di-CQA, which depended on the cultivar and leaf growth stage. The amount of phenolic acids in our own leaf samples harvested at BBCH stage 14 varied from 148.2 to 13.113 mg/100 g^−1^ DM. The lowest value was observed in the White Triumph cultivar, and the highest was observed in Okinava, which was also characterised by the highest content of flavonols in leaves harvested at the same growth stage. The second harvest round of sweet potato leaves took place at BBCH stage 51 (formation of side shoots), and the results of phenolic acid concentration ranged from 87.74 mg/100 g^−1^ DM in Molokai leaves to 4711 mg/100 g^−1^ DM in Radiosa leaves. The final harvest round of sweet potato leaves took place at BBCH stage 89 when they were fully ripe, and the concentration of phenolic acids ranged from 3649 mg/100 g^−1^ DM in Purple leaves to 10.168 mg/100 g^−1^ DM in Radiosa leaf samples. The highest concentration of phenolic acids was found in leaves harvested at full ripeness, i.e., at BBCH stage 89. A similar relation was found in a research study by Jaakola et al. [34], with the highest level of flavonoid compounds in the oldest leaves, which was probably caused by longer exposure to UV radiation. The earlier results of Jaakola et al. [34] were corroborated in a research study by Carvalho et al. [35], pointing to the higher concentration of polyphenolic compounds in leaves exposed to longer UV radiation (16 h) as compared to leaves exposed to UV light only for 8 h. Padda and Picha [24] and Jenga et al. [29] studied leaves of various sweet potato cultivars at different growth stages and found a significantly higher concentration of phenolic acids in young leaves; however, this was not confirmed in our own research, in which the highest concentration of polyphenolic compounds was recorded in leaves harvested at BBCH stage 89. It must be added that the discrepancies between the results can be caused by cultivar differentiation and cultivation in different climate conditions with lower temperatures. Research by Islam et al. [36] confirmed that high exposure to the sun and lower temperature during vegetation is conducive to higher levels of phenolic acids in sweet potato leaves.

As in the case of phenolic acids, the concentration of the measured flavonoids was differentiated by the physiological age of the test material and the genetic properties of the cultivars under study. In the first harvest round, at BBCH stage 14, the flavonol content in the sweet potato leaf samples was very differentiated. Nevertheless, in most cases, the measured flavonol content was below the detection limit. Flavonoid compounds were only detected in the Okinava, Beauregard, and Jewel cultivars. Their concentration ranged from 26.92 mg/100 g^−1^ DM in Jewel leaves to 925.3 mg/100 g^−1^ DM in Okinava leaves, as shown in Table 2. At BBCH growth stage 51, the presence of flavonoids was found in the leaves of all cultivars under study. The concentration of flavonoids varied from 46.48 mg/100 g^−1^ DM in the White Triumph leaf material to 315.7 mg/100 g^−1^ DM in Radiosa leaves. A much higher flavonol concentration was measured at full ripeness (BBCH 89), ranging from 413.2 mg/100 g^−1^ DM in White Triumph leaves to 1086 mg/100 g^−1^ DM in Carmen Rubin leaves. The results of our own study indicate a significant differentiation of flavonol content in sweet potato leaves depending on the cultivar. This was confirmed in a research study by Ojong et al. [37], which points to the significant differentiation of flavonol compound content in sweet potato leaves depending on the cultivar. The statistical analysis based on the Anova test showed a statistically significant differentiation of phenolic acid and flavonol content depending on the harvest date in the leaves of all cultivars under study. The agglomeration of data on the spatial structure of total polyphenolic compounds in the leaves of the cultivars under study with multivariate statistical analysis enabled the formation of groups characterised by common variety. The new data structure was created on the basis of the analysis of clusters and principal components presented in Figure 2, Figure 3 and Figure 4, which contain a graphic interpretation of the new data structure. Figure 2a,b presents an aggregation of polyphenolic compound concentration in the leaves under study at the first growth stage (BBCH 14)

Intergroup relations reveal five main clusters characterised by shared variance. Cluster 1 contains data describing the polyphenolic compound concentration in Jewel and Carmen Rubin leaves; cluster 2 contains the polyphenolic compound concentration in Beauregard and Radiosa cultivars. Cluster 3 includes the polyphenolic compound concentration in White Triumph and Purple leaves, whereas cluster 4 includes the aggregation of variables describing the total polyphenolic compounds in Molokai and Satsumo Imo cultivars. Cluster 5 contains a single element referring to the total polyphenols in Okinava leaves. Cluster 3 can be considered a formation characterised by a concentration of total polyphenols in leaves at the lowest average level of 265.6 mg/100 g^−1^ DM. Cluster 3 is most related to cluster 4 with an average level of polyphenolic compounds in the samples under study amounting to 881.4 mg/100 g^−1^ DM. The aggregation of factors making up cluster 1 contained polyphenolic compounds from sweet potato leaves at an average level of 1515 mg/100 g^−1^ DM. The concentration of polyphenolic compounds included in cluster 2 (polyphenols in Beauregard and Radiosa leaves) was 13 times higher than the polyphenol level in cluster 3, on average amounting to 3509 mg/100 g^−1^ DM. The single-element cluster 5 shows an exceptionally high concentration of polyphenolic compounds in Okinava leaves amounting to 14.038 mg/100 g^−1^ DM. The data structure shaping the concentration of polyphenolic compounds in the leaves of 9 sweet potato cultivars at BBCH growth stage 51 enables their aggregation. Based on cluster analysis and principal component analysis (PCA), we created a spatial image of the variable system, which is presented in Figure 3a,b.

The obtained data structure points to four areas of shared data variance. The space of the first area was filled with the concentration of polyphenols in Okinava, Purple, and Beauregard leaves. The cluster of the second area was formed by data describing the polyphenolic compound concentrations in leaves of cultivars with a similar polyphenolic compound concentration, including the following cultivars: Carmen Rubin, Jewel, and White Triumph. The polyphenol content in Molokai and Satsumo Imo leaves was included in cluster 3. Cluster 4 is a single-element subset that contains the concentration of polyphenolic compounds in the Radiosa leaves. Sweet potato leaves harvested at BBCH stage 51 demonstrated a varied concentration of polyphenolic compounds. The leaves of the Radiosa, White Triumph, Purple, and Beauregard cultivars demonstrated a statistically significant increase in the level of polyphenolic compounds as compared to the average concentration of polyphenols in the leaves of the 9 sweet potato cultivars under study, whereas the leaves of Okinava, Carmen Rubin, White Triumph, Molokai, Jewel, and Satsumo Imo were characterised by a statistically significant decrease of the polyphenol content as compared to samples collected at BBCH stage 14. The average lowest measured content of polyphenolic compounds in leaves at BBCH stage 51 was located in cluster 3, which included Molokai and Satsumo Imo leaves, with their level similar to White Triumph and Purple leaves harvested at BBCH stage 14 and amounting to 248 mg/100 g^−1^ DM. Cluster 2 presents a group of average polyphenol concentration of 821 mg/100 g^−1^ DM, including Carmen Rubin, Jewel, and White Triumph leaves, with an average measured content of the compounds that was 3.3 times higher than that in cluster 3. Data exploration of the polyphenol concentration in Okinava, Purple, and Beauregard leaves formed cluster 1, with the average polyphenolic compound content of 1905 mg/100 g^−1^ DM, which is 7.7 times higher than the polyphenol concentration in cluster 3. Clearly, as compared to the other cultivars, Radiosa leaves had the highest concentration of polyphenols amounting to 5027 mg/100 g^−1^ DM, which is 20 times higher than the concentration of polyphenolic compounds in cluster 3. Sweet potato leaves harvested at full ripeness, that is at BBCH stage 89, contained an average concentration of polyphenols of 6779 mg/100 g^−1^ DM, which in comparison to previous harvest rounds that involved an increase in the polyphenolic compound content by 56.76% and 77.54% for BBCH growth stages 14 and 51, respectively. Statistical analysis of our own research results based on cluster analysis and principal component analysis enabled the categorisation of three main agglomerations marked on Figure 4a,b as clusters 1, 2, and 3.

Cluster 1, with an average level of polyphenolic compounds of 4691 mg/100 g^−1^ DM, included Purple, White Triumph, and Molokai leaves. Data aggregation describing the coessentiality of data contained in cluster 2 with an average polyphenolic compound content of 6613 mg/100 g^−1^ DM, included Satsumo Imo, Okinava, Carmen Rubin, and Jewel leaves. Variable aggregation describing the polyphenolic compound concentration in Radiosa and Beauregard leaves formed cluster 1, with an average concentration of polyphenols of 10.243 mg/100 g^−1^ DM.

## 3. Materials and Methods

### 3.1. Reagents and Standards

Acetonitrile, acetic acid, methanol, ABTS (2,2’-azino-bis(3-ethylbenzothiazoline-6-sulfonic acid), Trolox (6-hydroxy-2,5,7,8-tetramethylchroman-2-carboxylic acid), TPTZ (2,4,6-Tris(2-pyridyl)-s-triazine), and DPPH (2,2-Diphenyl-1-picrylhydrazyl) were purchased from Sigma-Aldrich (Steinheim, Germany). Chlorogenic acid, quercetin-3-*o*-glucoside, and quercetin-3-*o*-galactoside were purchased from Extrasynthese (Lyon, France).

### 3.2. Plant Materials

The research was based on a rigorous field experiment conducted with the use of a random blocks method between 2016–2018 in Żyznów, Poland (49°49′ N or 21°50′ E). The experiment was conducted in three replicates on defective wheat complex soil with slightly acidic pH (6.0–6.5 pH in KCl). In the subsequent research years, soils did not differ in terms of the content of organic matter or macronutrient absorption. The average content of organic matter ranged from 15.0 to 18.7 g·kg^−1^. The content of absorbable phosphorus was 68.6–110 mg·P·kg^−1^, potassium 139–149 mg·K·kg^−1^, and magnesium 50–55 mg·Mg·kg^−1^. Nitrogen fertilisation was used simultaneously to NPK fertilisation in the following amounts: 80 kg·N; 34.9 kg·P; 99.6 kg·K·ha^−1^, and 25 t·ha^−1^ of manure. Manure fertilisation was used in the autumn, whereas mineral fertilisation was used in the spring, prior to planting. The propagating material included rooted cuttings of sweet potato from in vitro propagation. They were planted with 40 × 75 cm spacing in mid-May. The size of crop plots was 15 m^2^. During vegetation, cultivation was carried out in accordance with normal agricultural practice. The experiment had a two-factorial design. Factor 1 included sweet potato growth stages—their codes followed the BBCH scale (Biologische Bundesanstalt, Bundessortenamt und Chemische Industrie) [38]: main stem growth (stage 14), side shoot growth (stage 51), full ripeness (stage 89). Factor 2 included sweet potato cultivars (Okinava, Carmen Rubin, Radiosa, White Triumph, Molokai, Purple, Beauregard, Jewel, and Satsumo Imo). After each leaf harvest, the leaves were dried until solid in the lab dryer Alpha 1–2 LD plus.

### 3.3. Extraction Procedure

The finely powdered samples (1 g) were extracted with 10 mL of methanol acidified with 1% acetic acid. The extraction was performed twice by incubation for 20 min under sonication and with occasional shaking. Next, the slurry was centrifuged at 10,000 rpm for 10 min, and the supernatant was filtered through a 0.45-μm membrane and used for further chromatographic analysis.

### 3.4. Determination of Polyphenolic Compounds

Polyphenolic compounds were analysed using an ultra-performance reverse-phase liquid chromatography (UPLC)-PDA-MS/MS Waters ACQUITY system (Waters, Milford, MA, USA), consisting of a binary pump manager, sample manager, column manager, photodiode array (PDA) detector, and tandem quadrupole mass spectrometer (TQD) with electrospray ionisation (ESI). The separation was carried out using a BEH C18 column (100 mm × 2.1 mm i.d., 1.7 µm, Waters USA) kept at 50 °C. For the anthocyanins investigation, the following solvent system was applied: mobile phase A (2% formic acid in water *v*/*v*) and mobile phase B (2% formic acid in 40% ACN in water *v*/*v*). For other polyphenolic compounds, a lower concentration of formic acid was used (0.1% *v*/*v*). The gradient program was set as follows: 0 min 5% B, from 0 to 8 min linear to 100% B, and from 8 to 9.5 min for washing and back to initial conditions. The injection volume of the samples was 5 µL (partial loop with needle overfill), and the flow rate was 0.35 mL/min. The following parameters were used for TQD: capillary voltage, 3.5 kV; con voltage, 30 V in positive and negative mode; the source was kept at 120 °C, and the desolvation temperature was 350 °C; con gas flow, 100 L/h; and desolvation gas flow, 800 L/h. Argon was used as the collision gas at a flow rate of 0.3 mL/min. The polyphenolic detection and identification were based on a specific PDA spectra, mass-to-charge ratio, and fragment ions obtained after collision-induced dissociation (CID). Quantification was achieved by the injection of solutions of known concentrations ranging from 0.05 to 5 mg/mL of phenolic compounds as standards. All determinations were performed in triplicate and expressed as mg/L. Waters MassLynx software v.4.1 was used for data acquisition and processing.

### 3.5. Ferric-Reducing/Antioxidant Power Assay

The total antioxidant potential of the samples under study was measured with the FRAP method based on the reducing power of the compound (aqueous solution of iron salt) as the indicator of the antioxidant power [39]. A potential antioxidant reduces a ferric ion (Fe^3+^) to a ferrous ion (Fe^2+^/TPTZ), forming a blue complex created as a result of antioxidants giving up electrons at pH 3.6. The reaction is monitored by absorbance assay at 593 nm. The iron-reducing reagent antioxidant (FRAP) was obtained by mixing 300 mM acetate buffer and 10 mL of TPTZ in 40 mM HCl and 20 mM FeCl3.6H_2_O with a 10:1:1 ratio (*v*/*v*/*v*) at 37°. The freshly prepared active FRAP reagent was pipetted with a 1–5 mL variable micropipette (3.995 mL), mixed with 5 μL of appropriately diluted plant sample, and thoroughly mixed. An intense blue colour appeared when the iron–tripyridyltriazine (Fe^3+^ TPTZ) complex was reduced to a ferrous form (Fe^2+^), and absorbance at 593 nm was recorded in comparison to a blank feed of the reagent (3.995 mL of reagent FRAP + 5 μL distilled water) after 30 min incubation at 37°. All measurements were taken in three replicates. The standard curve was delineated with the use of various Trolox concentrations. All solutions were used on the day they were prepared. The results were corrected to account for dilution and expressed in µmol of Trolox equivalent per 100 g DM of the sample. The tests were carried out by means of the Thermo EVO 300PC Spectrophotometer (Thermo, Waltham, MA, USA).

### 3.6. Free Radical Scavenging Ability Determination Using a Stable ABTS Radical Cation

The free radical scavenging ability in plant samples was determined by means of the ABTS radical cation decolourisation assay [40]. An ABTS ^+^ radical cation was obtained in a reaction between 7 mM ABTS·^+^ in water and 2.45 mM potassium persulphate (1:1) stored in darkness at room temperature for 12–16 h prior to use. Then, the ABTS ^+^ solution was diluted with methanol until an absorbance of 0.700 at 734 nm was obtained. After adding 5 μL of plant extract to 3.995 mL of the diluted ABTS·^+^ solution, absorbance was measured after 10 min from initial mixing. Each assay included an appropriate blind solvent. For all analyses, a standard curve was drawn based on various Trolox concentrations. All measurements were taken in three replicates with the use of the Thermo EVO 300PC Spectrophotometer (USA). The results were corrected to account for dilution and expressed in µmol of Trolox per 100 g of DM.

### 3.7. DPPH

The total ability to scavenge free radicals in the sweet potato leaf samples under study was estimated on the basis of a slightly modified method by [41] with the use of a stable DPPH free radical with a maximum absorption at 515 nm. The radical solution was prepared by dissolving 2.4 mg DPPH in 100 mL of methanol. The test solution (5 μL) was added to 3.995 mL methanolic DPPH. The mixture was shaken well and kept in darkness at room temperature for 10 min. Absorbance of the reaction mixture was measured spectrophotometrically at 515 nm. In addition, absorbance of free radical DPPH without antioxidant was measured, i.e., in a blind feed. DPPH ^•^ tests were conducted in three replicates. The results were corrected to account for dilution and calculated on the basis of a curve formed from various Trolox concentrations, and the end result was expressed in µm of Trolox per 100 g DM.

### 3.8. Statistical Analysis

The results were presented as an average of three independent measurements. All analyses were carried out in the Statistica v. 13.3 software (StatSoft, Kraków, Poland). The significance of averages was tested with a post-hoc procedure in Duncan test, which is a part of one-way analysis of variance (ANOVA). Correlation analysis was conducted with Pearson two-way correlation. Differences of *p* < 0.05 were considered significant. In order to describe mutual relations between test objects, variable reduction techniques were applied: PCA and cluster analysis.

## 4. Conclusions

The leaves of nine sweet potato varieties grown in a temperate climate in south-eastern Poland contained 7 polyphenolic compounds, including 5 chlorogenic acids (5-CQA, 3-CQA, 4-CQA, 3,4-CQA, 3,5-CQA) and 2 flavonoids (Q3GA, Q3GL). Their content depended on the genetic properties of the cultivars under study and the growth stage of the leaves harvested at various BBCH-scale growth stages: 14, 51, and 89. The average content of the identified polyphenolic compounds in the varieties under study ranged from 148.2 mg/100 g^−1^ DM (White Triumph) to 14.038.6 mg/100 g^−1^ DM (Okinava) for leaves harvested at BBCH stage 14. In the case of leaves harvested at BBCH stage 51, the polyphenolic compound concentration ranged from 1500.1 mg/100 g^−1^ DM (Okinava) to 5026.8 mg/100 g^−1^ DM (Radiosa), whereas at BBCH stage 89, the results varied from 4078.1 mg/100 g^−1^ DM (Purple) to 11.183.5 mg/100 g^−1^ DM (Radiosa). Carmen Rubin leaves harvested at growth stage 14 were characterised by the highest content of polyphenolic compounds, whereas Okinava leaves had the highest amount of said compounds at growth stage 51. The highest level of polyphenolic compounds in leaves at the last growth stage, i.e., BBCH 89, was found in the Radiosa cultivar. The average highest activity of bioactive compounds was found in leaves harvested at stage 89 according to the BBCH scale. The highest concentration levels were found for 3-CQA at all BBCH growth stages. In most cases, significant correlations were determined between the content of polyphenols and antioxidant activity measured by means of ABTS, DPPH, and FRAP assays. The research confirms that sweet potato leaves are a source of bioactive substances; however, their content is strongly influenced by the genetic properties of the varieties and leaf development phases. The conducted studies contribute to a general understanding of the influence of leaf development phases and genotype on the accumulation of polyphenolic compounds. The results are innovative and can have a practical application, as the knowledge of the content of the substances under study makes it possible to determine the optimal management practice of sweet potato leaf harvest in order to obtain more top-quality raw material.

## Figures and Tables

**Figure 1 molecules-25-03473-f001:**
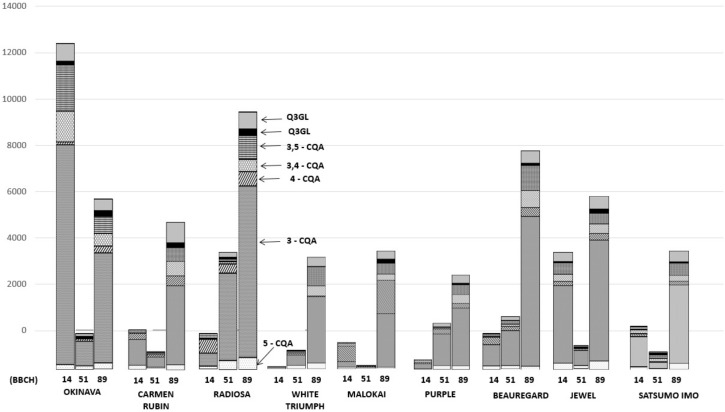
Polyphenol content in sweet potato leaves (mg/100 g^−1^ DM) depending on the development phase according to the BBCH scale.

**Figure 2 molecules-25-03473-f002:**
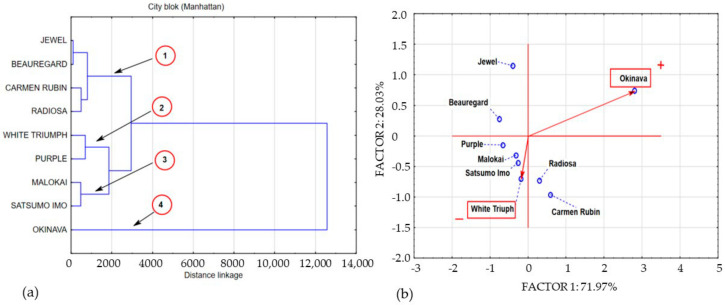
Dendrogram and bio-plot (PCA) of average concentrations of polyphenols in sweet potato leaves collected in BBCH phase 14. (**a**) 1,2,3,4 clusters. (**b**) + variety with the highest concentration polyphenolics compounds, − variety with the lowest concentration polyphenolics compounds.

**Figure 3 molecules-25-03473-f003:**
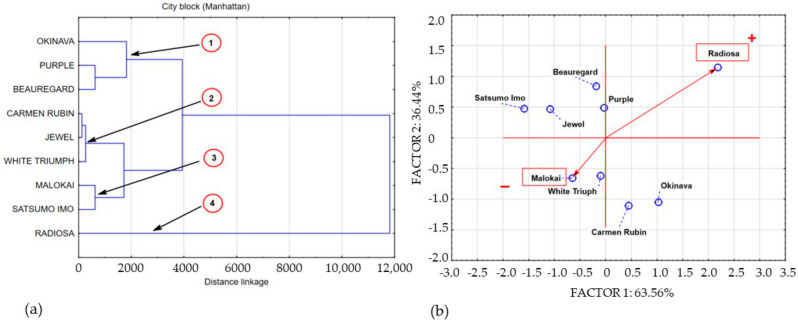
Dendrogram and bio-plot (PCA) of average concentrations of polyphenols in sweet potato leaves collected in BBCH phase 51. (**a**) 1,2,3,4 clusters. (**b**) + variety with the highest concentration polyphenolics compounds, − variety with the lowest concentration polyphenolics compounds.

**Figure 4 molecules-25-03473-f004:**
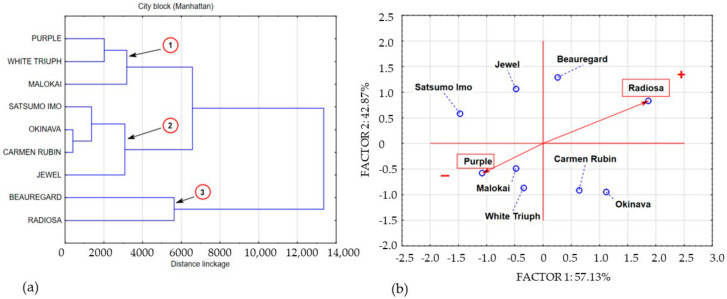
Dendrogram and bio-plot (PCA) of average concentrations of polyphenols in sweet potato leaves collected in BBCH phase 89. (**a**) 1,2,3,4 clusters. (**b**) + variety with the highest concentration polyphenolics compounds, − variety with the lowest concentration polyphenolics compounds.

**Table 1 molecules-25-03473-t001:** Characterisation of phenolics compounds from sweet potato extracts collected from the HPLC column and analysed by UV-VIS, HPLC, and LC/MS. 3-CQA: chlorogenic acid, 5-CQA: neochlorogenic acid, 4-CQA: 4-cryptochlorogenic acid, 3,4-diCQA: 3,4-di-*O*-caffeoylqunic acid, 3,5-diCQA: 3,5-di-*O*-caffeoylqunic acid.

Peak No.	Rt (min)	UV-Vis nm	[M−H]^−^ (*m*/*z*)	Fragment Ion (*m*/*z*)	Compounds
phenolics acid					
1	2.335	240/298/327	353	191,179,135	3-CQA
2	2.960	241/298/328	353	191	5-CQA
3	3.139	244/298/326	353	173,179,191,135,155,161	4-CQA
6	5.153	240/298/326	515	135,155,161,173,179,191,353	3,4-diCQA
7	5.275	239/298/328	515	135,179,191,335,353	3,5-diCQA
flavonols					
4	4.727	256/265/354	463	151,179,255,271,300,301	Quercetin-3-*O*-galactoside
5	4.763	256/265/354	463	151,179,255,271,300,301	Quercetin-3-*O*-glucoside

**Table 2 molecules-25-03473-t002:** Antioxidant activity and correlation coefficients for sweet potato (*Ipomoea batatas* L. [Lam.]) leaf extracts measured by tests ABTS ^o+^, FRAP and DPPH [µmol TE (Trolox)/100 g^-1^ DM.].

Varieties	Okinava	Carmen Rubin	Radiosa	White Triumph	Malokai	Purple	Beauregard	Jewel	Satsumo Imo
Harvest time in phase 14 according to the BBCH scale									
**Polyphenols** **mg · 100 g^−1^ DM.**	**14.0386**	**1767.8**	**1604.9**	**148.2**	**962.1**	**383.1**	**1426.8**	**5251**	**800.91**
ABTS	138.16	52.67	51.44	24.24	30.03	30.82	41.45	90.53	26.97
Correlation coefficient	**−0.838**	**−0.994**	**−0.600**	**−0.997**	**0.999**	**0.993**	**0.995**	**0.675**	**0.141**
DPPH	190.75	46.63	49.75	18.55	22.73	28.17	39.93	117.46	28.49
Correlation coefficient	**−0.958**	**0.755**	**0.379**	**0.999**	**0.978**	**0.999**	**0.931**	**0.711**	**−0.362**
FRAP	551	144.5	147.3	52.8	72.6	95.4	115.6	326.3	86.4
Correlation coefficient	**−0.974**	**0.839**	**−0.286**	**0.997**	**0.986**	**0.999**	**0.999**	**0.597**	**−0.565**
Harvest time in phase 51 according to the BBCH scale									
**Polyphenols** **mg · 100 g^−1^ DM.**	**1500.8**	**829.2**	**5026.8**	**760.8**	**144.76**	**2003.6**	**2211.3**	**873.1**	**351.35**
ABTS	41.49	27.72	63.39	26.48	26.49	11.4	55.77	30.52	12.44
Correlation coefficient	**0.705**	**0.998**	**0.993**	**0.664**	**0.664**	**0.999**	**−0.395**	**0.999**	**−0.394**
DPPH	105.64	50.59	187.39	47.6	47.60	10.47	122.56	58.63	38.66
Correlation coefficient	**0.658**	**0.987**	**0.974**	**−0.956**	**−0.956**	**0.998**	**−0.369**	**0.999**	**−0.719**
FRAP	283.46	172.56	617.43	138.5	138.58	28.47	415.43	168.45	131.34
Correlation coefficient	**0.537**	**0.999**	**0.996**	**0.999**	**0.999**	**0.999**	**−0.381**	**0.998**	**−0.772**
Harvest time in phase 89 according to the BBCH scale									
**Polyphenols** **mg · 100 g^−1^ DM.**	**7059.6**	**6333.4**	**11.1835**	**4802.4**	**5193.3**	**4078.1**	**9303.5**	**7731.2**	**5326.5**
ABTS	93.35	108.6	126.52	112.65	90.62	101.52	113.6	108.6	89.46
Correlation coefficient	**0.888**	**0.537**	**0.805**	**0.301**	**0.756**	**0.776**	**−0.141**	**0.980**	**0.492**
DPPH	110.62	145.6	197.43	124.81	114.59	109.72	171.7	147.3	65.77
Correlation coefficient	**0.656**	**0.785**	**0.813**	**0.227**	**0.776**	**0.665**	**−0.352**	**0.954**	**0.549**
FRAP	308.0	455.3	636.31	382.0	385.0	334.42	558.4	462.4	195.61
Correlation coefficient	**0.817**	**−0.678**	**0.367**	**0.327**	**0.720**	**0.773**	**−0.272**	**0.962**	**−0.201**

**Table 3 molecules-25-03473-t003:** Polyphenols content in sweet potato leaves (mg 100 g^−1^ DM) depending on the development phase according to the BBCH (Biologische Bundesanstalt, Bundessortenamt und Chemische Industrie) scale.

Varieties	BBCH	Phenolics Acid					Sum	Flavonoids			Total
		5-CQA	3-CQA	4-CQA	3.4-diCQA	3.5-diCQA		Q3GA	Q3GL	Sum	
Okinava	14	223.3	9455	133.0	1322	1980	13,113	173.8	751.5	925.3	14,038.6
	51	148.6	1023	112.0	15.65	50.95	1350	30.18	119.7	149.9	1500.1
	89	250.3	4572	272	522.3	713	6330	235.2	494.8	730.0	7059.6
Carmen Rubin	14	184.8	1160	271.4	2.09	149.5	1768	*nt*	*nt*	*nt*	1767.8
	51	52.2	507.0	113.8	49.61	44.51	767.1	23.21	38.87	62.08	829.2
	89	224	3386	426.8	633.2	577.0	5248	210.6	875.8	1086	6333.4
Radiosa	14	140.27	577.9	621.3	59.39	206.0	1604.9	*nt*	*nt*	*nt*	1604.9
	51	403.79	3711	402.5	73.6	120.2	4711	63.27	252.4	315.7	5026.8
	89	527.44	7451	611.9	540.8	1037	10,168	269.6	745.8	1015	11,183.5
White Triumph	14	*nt*	100	*nt*	*nt*	48.19	148.2	*nt*	*nt*	*nt*	148.2
	51	135.8	416.9	65.44	45.27	50.87	714.3	39.63	6.85	46.48	760.8
	89	245.1	2855	35.98	429.9	823.2	4389	18.69	394.5	413.2	4802.4
Malokai	14	63.69	200.5	560.3	77.55	60.06	962.0	*nt*	*nt*	*nt*	962.1
	51	*nt*	59.64	28.1	*nt*	*nt*	87.74	37.97	19.05	57.03	144.76
	89	51.26	2371	1475	265.7	479.6	4643	195.7	355.0	550.7	5193.3
Purple	14	20.95	192.5	54.87	69.06	45.7	383.1	*nt*	*nt*	*nt*	383.1
	51	160.2^BC^	1372	215.5	32.58	40.61	1821	20.57	162.1	182.7	2003.6
	89	158.4^CB^	2490	205.7	379.4	415.8	3649	77.59	351.6	429.2	4078.1
Beauregard	14	145.4^AB^	832.5	303.9	22.22	95.89	1400	*nt*	26.92	26.92	1426.8
	51	146.5^BA^	1477	176.8	89.41	155.6	2046	3.41	162.6	166.0	2211.3
	89	156.6	6364	372.7	731.4	1060	8685	94.17	524.6	618.8	9303.5
Jewel	14	285.09	3473	193.2	323.2	506.7	4781	66.42	403.4	469.8	5251
	51	109.45	578.9	65.16	30.81	29.07	813.4	3.37	56.35	59.71	873.1
	89	379.02	5399	283.7	425.0	481.4	6969	190.0	573.1	763.1	7732.2
Satsumo Imo	14	54.58	559.5	54.8	73.12	58.91	800.9	*nt*	*nt*	*nt*	800.91
	51	3.93	131.34	11.24	71.99	72.66	291.2	39.93	20.26	60.19	351.35
	89	261.6	3527	175.0	261.5	541.2	4766	74.97	485.2	560.2	5326.5

5-CQA: neochlorogenic acid, 3-CQA: chlorogenic acid, 4-CQA: 4-cryptochlorogenic acid, 3,4-diCQA: 3,4-di-*O*-caffeoylqunic acid, 3,5-diCQA: 3,5-di-*O*-caffeoylqunic acid, Q3GA: quercetin-3-*O*-galactoside, Q3GL: quercetin-3-*O*-glucoside. Statistically not significant differences were marked with letters in the varieties A (BBCH 14), B (BBCH 51), C (BBCH 89). *nt*—not detected.

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
