# Peer review of "The Content of Phenolic Acids and Flavonols in the Leaves of Nine Varieties of Sweet Potatoes (Ipomoea batatas L.) Depending on Their Development, Grown in Central Europe"

_molecules, 2020, doi:10.3390/molecules25153473_

Round 1

Reviewer 1 Report

Comments are attached

Author Response

Response to Reviewer 1 Comments

Manuscript ID: molecules-882383

Title: The content of phenolic acids and flavonols in the leaves of nine
varieties of sweet potatoes (Ipomoea batatas L.) depending on their
development, grown in Central Europe

Reviewer 1:

Thank you for the remarks and suggestions of Reviewer 1. The manuscript has been corrected according to the remarks and suggestions of Reviewer 1. 

Point 1.Why were phenolic acids and flavonoids measured in leaves and not in potatoes directly?

Response 1: The food raw material of sweet potato are tubers and leaves.  The content of active ingredients in sweet potato leaves has been researched in a number of scientific studies, whose aim was to determine the health benefit of the leaves. However, few studies have attempted to determine the influence of leaf growth stages and genotype of sweet potato cultivars on the content of phenolic acids and flavonoids. Therefore, we decided to conduct such a study, and the results have shown that the growth stages and genetic properties of cultivars have a significant influence on the content of phenolic acids and flavonoids in sweet potato leaves. The results are innovative and can have a practical application, as the knowledge of the content of the substances under study makes it possible to determine the optimal management practice of sweet potato leaf harvest in order to obtain more top-quality raw material.               However, considering the suggestion of Reviewer 1, the next stage of the study will be to determine the content of phenolic acids and flavonoids in sweet potato tubers depending on the growth stage and cultivar.

Point 2. L57 Please see whether aforementioned is only a word. Check the posible typo.

Response 2: L 57 Thank you for the remark, the word “aforementioned” in the text was replaced with “mentioned above”.

Point 3. L84 Please check the secuence of the identified compounds dispalyed in Table 1 (1, 2, 3, 6, 7, 4, 3)

Response 3: L84 Dear Reviewer, let us clear the ambiguities: The numbering in Table 1 is correct, because Table 1 presents the characteristics of the identified polyphenolic compounds (phenolic acids, flavonoids). Column 1 contains the numbers of chromatography files reflected in retention times in column 2. In the table, compounds are pooled according to their group, i.e. phenolic acids and flavonoids (column 6).

Point 4. L87, 96, Change to m/z in cursive

Response 4: L87, 96 Thank you for the remark, m/z was written in cursive as m/z in the entire text. 

 Point 5. L123, 133 Please provide P values for the Pearson correlation moments

Response 5: Thank you for the remark, L 123, L 133 were corrected to contain the necessary information.

L123 is : (Pearson correlation = -0.0319, -0.369, -0.381, respectively)Corrected to:L 123 (Pearson Correlation coefficient, R2= -0.0319, R2= -0.369, R2= -0.381, respectively) L 133 is : A similar high correlation coefficient of 0.98 between antioxidant activity measured with the ABTS and DPPH method was arrived at by Zhang et al. [21] and Fu et al. [28].Corrected to: A similar high correlation coefficient of R2= 0.98 between antioxidant activity measured with the ABTS and DPPH method was arrived at by Zhang et al. [21] and Fu et al. [28].

Reviewer 2 Report

The authors have presented their work entitle "The content of phenolic acids and flavonols in the leaves of nine varieties of sweet potatoes (Ipomoea batatas L.) depending on their development, grown in Central Europe ". The topic is relevant. Therefore, I recommend the article for publication after some MINOR REVISIONS.

My suggestions are:

Line 23: 14038.6 mg.100g-1 DM

Please, the number 100 could be separate to g-1, same in line 25 and 26.

Other question, what is DM? you could be define this acronym.

The authors use analise and analize, please unify in the whole manuscript.

Table 2, first column “Vartieties” please, correct this word.

Line 158: “nt: not detected” in the table 2, authors used “nd” not “nt”, please, modify.

Line 160: “Figure 1. Polyphenol content in sweet potato leaves (mg. 100 g-1 d.m) depending on the development phase according to the BBCH scale.” All text the authors put “DM” here d.m please, unify.

Lines 163 and 164, also line 177, and 179: the number 100 could be separate to g-1 Please, revise all the document.

Author Response

Responses to comments and suggestions of the Reviewer 2

Manuscript ID: molecules-882383

Title: The content of phenolic acids and flavonols in the leaves of nine
varieties of sweet potatoes (Ipomoea batatas L.) depending on their
development, grown in Central Europe

Reviewer 2:

Thank you for the Reviewer’s remarks and suggestions. The manuscript has been corrected according to all of the remarks and suggestions of Reviewer 2. 

Point 1: Line 23: 14038.6 mg.100g-1 DM

Response 1: Line 23 Thank you for the remark, corrected to 14038.6 mg.100 g-1 DM.

Point 2: Please, the number 100 could be separate to g-1, same in line 25 and 26.

Response 2: Thank you for the remark, corrected as follows: in line 25 and 26 – from 144.76 to 5026.8 mg.100 g-1 DM, and at BBCH stage 89 – from 4078.1 to 11183.5 mg.100 g-1 DM.

Point 3: Other question, what is DM? you could be define this acronym.

Response 3: DM is short for dry mass, which is the measurement of completely dried objects.  

Point 4: The authors use analise and analize, please unify in the whole manuscript.

Response 4: Thank you for the remark, corrected to “analyse” in the text.

Point 5: Table 2, first column “Vartieties” please, correct this word.

Response 5: Thank you for the reviewer’s remark, “Vartieties” corrected to “Varieties”. 

Point 6: Line 158: “nt: not detected” in the table 2, authors used “nd” not “nt”, please, modify.

Response 6: Thank you for the reviewer’s remark, in Table 2 the erroneous “nd” was corrected to “nt”.

Point 7: Line 160: “Figure 1. Polyphenol content in sweet potato leaves (mg. 100 g-1 d.m) depending on the development phase according to the BBCH scale.” All text the authors put “DM” here d.m please, unify.

Response 7: Line 160 Thank you for the reviewer’s remark, the title was corrected to Figure 1 Polyphenol content in sweet potato leaves (mg.100 g-1 DM) depending on the development phase according to the BBCH scale. In the entire manuscript, d.m. was corrected to DM. 

Point 8: Lines 163 and 164, also line 177, and 179: the number 100 could be separate to g-1 Please, revise all the document.

Response 8: Thank you for the reviewer’s remark, it was corrected and “g” was separated.

Line 163 5026.8 mg.100 g-1 DM

Line 164 11183.5 mg.100 g-1 DM

Line 177 4200.9 mg.100 g-1 DM,

Line 179 13113 mg.100 g-1 DM

In the entire manuscript, “g” was corrected and separated.

The entire manuscript has been linguistically corrected.
